# Development of Novel Peptidyl Nitriles Targeting Rhodesain and Falcipain-2 for the Treatment of Sleeping Sickness and Malaria

**DOI:** 10.3390/ijms25084410

**Published:** 2024-04-17

**Authors:** Carla Di Chio, Josè Starvaggi, Noemi Totaro, Santo Previti, Benito Natale, Sandro Cosconati, Marta Bogacz, Tanja Schirmeister, Jenny Legac, Philip J. Rosenthal, Maria Zappalà, Roberta Ettari

**Affiliations:** 1Department of Chemical, Biological, Pharmaceutical and Environmental Sciences, University of Messina, Viale Ferdinando Stagno d’Alcontres, 98166 Messina, Italy; cdichio@unime.it (C.D.C.); starvaggi4@gmail.com (J.S.); noemi.totaro@studenti.unime.it (N.T.); spreviti@unime.it (S.P.); mzappala@unime.it (M.Z.); 2Department of Environmental, Biological and Pharmaceutical Sciences and Technologies, University of Campania Luigi Vanvitelli, Via Vivaldi 43, 81100 Caserta, Italy; benito.natale@unicampania.it (B.N.); sandro.cosconati@unicampania.it (S.C.); 3Institute of Organic Chemistry & Macromolecular Chemistry, Friedrich Schiller University of Jena, Humboldtstraße, 10, DE 07743 Jena, Germany; marta.bogacz@uni-jena.de; 4Institute of Pharmacy and Biochemistry, University of Mainz, Staudingerweg 5, DE 55128 Mainz, Germany; schirmei@uni-mainz.de; 5Department of Medicine, San Francisco General Hospital, University of California, 1001 Potrero Avenue, San Francisco, CA 94110, USA; jenny.legac@ucsf.edu (J.L.); philip.rosenthal@ucsf.edu (P.J.R.)

**Keywords:** neglected tropical diseases, poverty-related diseases, rhodesain and falcipain-2

## Abstract

In recent decades, neglected tropical diseases and poverty-related diseases have become a serious health problem worldwide. Among these pathologies, human African trypanosomiasis, and malaria present therapeutic problems due to the onset of resistance, toxicity problems and the limited spectrum of action. In this drug discovery process, rhodesain and falcipain-2, of *Trypanosoma brucei rhodesiense* and *Plasmodium falciparum*, are currently considered the most promising targets for the development of novel antitrypanosomal and antiplasmodial agents, respectively. Therefore, in our study we identified a novel lead-like compound, i.e., inhibitor **2b**, which we proved to be active against both targets, with a *K*_i_ = 5.06 µM towards rhodesain and an IC_50_ = 40.43 µM against falcipain-2.

## 1. Introduction

Neglected tropical diseases (NTDs) and poverty-related diseases (PRDs) include a group of infections predominately occurring in tropical areas affecting the most disadvantaged communities, in particular, women and children [1]. One of the most important NTDs is human African trypanosomiasis (HAT) [2], while malaria is the most widespread PRD in the world [3].

HAT (also known as sleeping sickness) is a parasitic disease widespread in sub-Saharan Africa, caused by trypanosome parasites, and transmitted by tsetse fly of *Glossina* genus. Two subspecies of *Trypanosoma brucei* cause this disease: *T. b. gambiense* in West Africa and *T. b. rhodesiense* in East Africa [2].

HAT is characterized by two main stages: in the first stage, also known as the hemolymphatic stage, a bloodstream invasion by the parasite induces fever and muscle aches; if untreated, the first stage can evolve into the neurological one, in which the penetration of the central nervous system by trypanosomes induces neurological disturbances, sleep disorders, and finally death. Malaria is transmitted through the bites of infected female mosquitoes of the *Anopheles* genus. Five Plasmodium parasite species cause malaria in humans. In particular, *P. falciparum* is the most dangerous and lethal species; it predominates in sub-Saharan Africa and is common in other tropical regions. *P. vivax* is the dominant malaria parasite in many countries outside of sub-Saharan Africa. A large number of other malaria species can infect humans; some of them are *P. malariae*, *P. ovale*, and *P. knowlesi* [3].

The most common symptoms of malaria are fever, chills, and headache. According to the latest World Malaria Report from the World Health Organization (WHO), there were an estimated 249 million cases of malaria in 2022, leading to 608,000 deaths in 2023 [3]. Available antitrypanosomal and antimalarial agents have multiple limitations. The initial HAT treatment included only four drugs, among which three, i.e., suramin, pentamidine, and melarsoprol, were developed more than 60 years ago. Suramin and pentamidine are active on the hemolymphatic stage of the disease, while melarsoprol and eflornithine act in the neurological stage [4].

Melarsoprol, the most toxic of these compounds, is a trivalent arsenical and, unfortunately, it causes reactive encephalopathy in 5–10% of treated patients, with 1–5% mortality [4], Nifurtimox, a 5-nitrofuran with antifungal properties, formally approved for the American trypanosomiasis, is used off-label, in combination with eflornithine, for the treatment of the gambiense trypanosomiasis (g-HAT) [5].

Almost all drugs are administered parenterally, so there is always the problem of compliance. Fexinidazole was recently introduced into therapy and orally administered; however, it was approved for g-HAT only, so for the most severe form of HAT, namely the rhodesiense form (r-HAT), apart from the toxic melarsoprol, currently there are no antytripanosomal drugs [6]. Artemisinin-based combination therapies (ACTs) are the mainstays for the treatment of uncomplicated malaria. This therapy includes two antimalarial agents with different mechanisms of action: artemether/lumefantrine, artesunate/amodiaquine (ASAQ), dihydroartemisinin/piperaquine, artesunate/pyronaridine (AS-pyronaridine), and artesunate/sulfadoxine/pyrimethamine. Severe malaria is treated with intravenous artesunate followed by a full course of an ACT once the patient can tolerate oral medicines [7].

Unfortunately, malaria remains an enormous problem, and its treatment and control are seriously challenged by the emergence of drug resistance. Partial resistance to artemisinins emerged in southeast Asia early this century and, more recently, emergence of different mutations that are validated markers of resistance has been documented in multiple places in eastern Africa, including Rwanda, Uganda, Tanzania, Ethiopia, and Eritrea [8].

In the present scenario, there is an urgent need to develop new drugs, ideally directed against novel targets, for the treatment of HAT and malaria. In this context, we focused our attention on inhibitors of the parasite cathepsin-L-like cysteine proteases rhodesain of *T. b. rhodesiense* and falcipain-2 (FP-2) of *P. falciparum*.

Rhodesain is a clan CA, family C1 (papain family) cysteine protease, essential for the parasite’s survival [9,10]. It is involved in parasite invasiveness, allowing it to cross the blood–brain barrier (BBB), leading to the neurological stage of HAT. To evade the host immune response, rhodesain takes part in the turnover of variant surface glycoproteins of the trypanosome coat and the degradation of human immunoglobulin [11,12].

FP-2 is a clan CA family C1 cysteine protease that hydrolyzes hemoglobin to provide amino acids that are essential to the parasite for protein synthesis. Moreover, FP-2 may also cleave off cytoskeletal proteins, i.e., ankyrin and band-4.1, to facilitate the rupture of the red-cell membrane. Both rhodesain and FP-2 contain a left (L) and a right (R) domain, with the catalytic triad (Cys/His/Asn) located in a cleft between the two domains [13,14]. Our research group has been involved in the development of novel rhodesain and FP-2 inhibitors for the treatment of HAT and malaria [15,16,17,18,19,20,21,22,23,24,25,26,27,28,29,30,31].

In the present study, we designed a new series of peptidyl nitrile-based inhibitors. A very active area of pharmaceutical research concerns the design of peptidomimetics as new therapeutic agents, as they show good stability against enzymatic degradation and good oral bioavailability. The use of conformationally constrained peptidomimetics should provide selectivity towards specific proteases.

## 2. Results and Discussion

### Chemistry

The designed peptidomimetics **1-4a-b** (Figure 1) are characterized by a nitrile group as an electrophilic warhead able to interact with the catalytic cysteine of the target proteases FP-2 and rhodesain, to give a reversible thioimidate adduct. At the P3 site, an unsubstituted indole or a 2- or 3-methyl or 5-fluoro substituted indole scaffold was introduced within a characteristic peptide sequence. The choice of this scaffold was based on consistent data showing that several inhibitors containing this nucleus had marked antiplasmodial activity [32]. At the P2 site, a glycine or a *β*-alanine residue was introduced to evaluate the optimal distance between the indole scaffold and the warhead for the accommodation in the protease pockets. At the P1 site, a homophenylalanine (hPhe) residue, a preferential amino acid for the recognition of the inhibitors by both FP-2 and rhodesain, was introduced. In addition, we also developed a second series of inhibitors, **5-8a-b** (Figure 1), bearing at the P1 site a cyclopropyl carbonitrile residue present in the macrolactam co-crystallized with rhodesain (PDB: 6EXO) [33], to evaluate the impact of this structural modification on rhodesain inhibition. In the present study, we report the synthesis of these new compounds, investigation of activities against rhodesain and FP-2, and docking studies.

The synthesis of the P3-P2 synthons was realized starting from differently substituted indole scaffolds [32], which were *N*-alkylated with ethyl-2-bromoacetate **13a** or ethyl-3-bromopropionate **13b** to obtain intermediates **14-17a-b**. Alkaline hydrolysis of the ester function afforded the corresponding carboxylic acids **18-21a-b** (Figure 1).

The final products **1-8a-b** were obtained via coupling reactions between the carboxylic acids **18-21a-b** and the (*S*)-2-amino-4-phenylbutanitrile, which was obtained according to the procedure that we recently reported [18], or the commercially available 1-aminocyclopropanocarbonitrile **23** (Figure 2).

All compounds **1-8a-b** were tested against recombinant rhodesain [34] by using Cbz-Phe-Arg-AMC as a fluorogenic substrate (Table 1). First, a preliminary screening at a fixed inhibitor concentration of 100 μM was performed. 

An equivalent volume of DMSO was used as the negative control, and E-64, an irreversible inhibitor of clan CA family C1 cysteine proteases, was used as the positive control. For compounds that inhibited the enzyme activity at least 50% at 10 μM, continuous assays were performed at seven different concentrations to determine dissociation constants.

The best inhibitory activity was obtained for compounds having the 2-methyl substituted indole nucleus at the P3 site and a hPhe residue at the P1 site (i.e., **2a** and **2b**; Table 1), which is the preferential amino acid for the recognition of the inhibitor by rhodesain. Compound **2b**, bearing a *β*-alanine residue at the P2 site, showed a slightly better enzymatic affinity (*K*_i_ = 5.06 µM) than the glycine derivative **2a** (*K*_i_ = 6.56 µM), as shown in Table 1.

Compound **6b** was also active, with a *K*_i_ value of 8.18 µM. This compound contains a 2-methyl substituted indole nucleus at the P3 site, a *β*-alanine at the P2 site, and a cyclopropryl carbonitrile residue at the P1 site present, which as previously mentioned, is present in the macrolactam inhibitor with which the rhodesain was co-crystallized.

In the case of compounds containing a hPhe residue at P1 and a glycine at P2, the order of reactivity was 2-methyl substituted indole (i.e., **2a**) > 3-methyl substituted indole (i.e., **3a**) > unsubstituted indole (i.e., **1a**), with *K*_i_ values of 6.56 µM, 10.31 µM, and 13.11 µM, respectively. In contrast, compounds with a 5-fluoro-indole were inactive. All compounds were also tested against FP-2, and in this case inhibitor **2b** was the only active compound, with an IC_50_ of 40.43 µM (Table 1). All the active compounds were tested against *T. b. brucei*, and the only active compound was inhibitor **2b**, with an IC_50_ of 12 µM at both 24 h and 48 h of incubation (Figure 2).

To gain deeper insight into the binding interactions of the newly discovered rhodesain inhibitors, molecular modeling studies were conducted. Specifically, docking experiments employed the X-ray structure of rhodesain bound to its inhibitor K11002 (PDB code: 2P86) [35]. Considering the ability of the inspected ligands to form a reversible covalent adduct with the enzyme, a covalent docking protocol was adopted, employing the flexible side-chain method available within the AutoDock4 (AD4, v4.2.6) docking software. We chose to include **2b** in this investigation, since it is the most active rhodesain inhibitor among those displaying inhibitory properties and antitrypanosomal activity (Table 1). Thus, **2b** was covalently docked into the X-ray structure of rhodesain after simulating a modified Pinner reaction between the SH group of residue C25 and the nitrile warhead [36], resulting in a thioimidate group responsible for the reversible covalent bond (Figure 3).

The ligand is represented as orange sticks. Rhodesain is represented as white sticks and ribbons, and a blue surface. H-bonds are represented as green dashed lines. Pocket locations are indicated by red labels.

In the lowest energy conformation predicted by AD4, the ligand was located in the enzyme binding site at a position resembling the experimental binding position of the co-crystallized ligand. Specifically, the hPhe residue was pointing toward the enzyme S1’ pocket with its aromatic ring positioned near H162 and W184 of the same pocket, where charge-transfer interactions can be established. The absence of the hPhe residue appeared to be detrimental for activity, with decreased rhodesain inhibition for compounds featuring the cyclopropyl moiety (compounds **5** to **8** in Table 1). In fact, the chemical nature of the latter did not permit it to efficiently contact the S1’ pocket. It is worth noting that the nature of the thioimidate group permitted the formation of a H-bond with the backbone NH of G163 residue G163. The CO group of the P2 residue contacted the backbone NH of G66, while the NH of the same residueP1 site formed H-bonds with the backbone CO of D161. The presence of two methylene groups in the P2 region of the ligand seemed to better orient the attached indole ring towards the S2 pocket, where it established hydrophobic interactions with L67, M68, and A138. Of note, it has been reported that A138 limits the space available in this pocket [10,35].

To further refine the predicted binding geometry, a 500 ns long molecular dynamics (MD) simulation was performed on the **2b** docked complex. The results of the MD simulation suggested a preferential binding orientation, with the adopted conformation of **2b** (Figure 4A) fairly stable throughout the simulation time, as demonstrated by the root mean square deviation (RMSD) calculated for the ligand’s atoms over time (Figure 4B).

The MD studies highlighted that **2b** can form a second H-bond with the backbone CO of D161 thanks to its thioimidate moiety, while losing the interaction with G66. Additionally, **2b** seemed to point its 2-methyl group towards nearby L67 and M68 in the S2 pocket. Interestingly, the role of the 2-methyl indole substitution was also underscored by the generally lower rhodesain inhibition of compounds devoid of this group (**1b**) or featuring the same group in position 3 of the same ring (**3b**).

## 3. Materials and Methods

### 3.1. Chemistry

All reagents and solvents were obtained from commercial suppliers and were used without any further purification. Melting points were determined in open capillary tubes on a Stuart Scientific melting point apparatus SMP3 and were uncorrected. To determine the purity of compounds elemental analyses were carried out on a C. Erba Model 1106 (Elemental Analyser for C, H, and N) instrument, and the obtained results were within ± 0.4% of the theoretical values. Merck silica gel 60 F254 plates were used for analytical TLC; flash column chromatography was performed on Merck silica gel (200–400 mesh). ^1^H and ^13^C and NMR spectra were recorded on a Varian 500 MHz spectrometer equipped with a ONE_NMR probe and operating at 499.74 and 125.73 MHz for ^1^H and ^13^C, respectively. We used the residual signal of the deuterated solvent as an internal standard. Splitting patterns are described as singlet (s), doublet (d), doublet of doublet (dd), triplet (t), quartet (q), multiplet (m), or broad singlet (bs). ^1^H and ^13^C NMR chemical shifts (δ) are expressed in ppm and coupling constants (*J*) are given in Hz. All ^1^H- and ^13^C-NMR spectra are reported in the Appendix A.


**Synthesis of P3-P2 synthons**


Ethyl 2-(1*H*-indol-1-yl) acetate **(14a)**

In a double-neck round bottom flask containing NaH 57% dispersion in mineral oil (1.62 g, 0.038 mol), placed in an ice bath at 0 °C, ^1^H-indole **9** (3 g, 0.026 mol), previously solubilized in anhydrous DMF, was added dropwise, under nitrogen atmosphere. The reaction mixture was stirred for 1 h and then ethyl-2-bromoacetate **13a** (4.26 mL, 0.038 mol) was added. The reaction mixture was stirred overnight at room temperature. After overnight stirring, the reaction mixture was quenched with a saturated NH_4_Cl solution at 0 °C; then, the mixture was extracted with EtOAc, washed with brine, dried over Na_2_SO_4_, filtered, and evaporated in vacuo. The residue was purified by silica gel column chromatography using CHCl_3_/Acetone 95:5 to obtain the pure product **14a** (1.74 g, 33%); consistency: light yellow oil; *R*_f_ = 0.43 (petroleum ether/EtOAc 9:1). ^1^H NMR (500 MHz, CDCl_3_) = δ:1.15 (t, *J* = 7.1 Hz, 3H), 4.10 (q, *J* = 7.1 Hz, 2H), 4.66 (s, 2H), 6.49 (d, *J* = 2.9 Hz, 1H), 6.96 (d, *J* = 2.9 Hz, 1H), 7.10–7.06 (m, 1H), 7.16 (d, *J* = 4.1 Hz, 2H), 7.59 (d, *J* = 7.9 Hz, 1H) ppm; ^13^C NMR (125 MHz, CDCl_3_) = δ: 13.89, 47.50, 61.34, 102.13, 108.77, 119.60, 120.86, 121.74, 128.35, 128.41, 136.28, 168.35. Elemental analysis: calcd for C_12_H_13_NO_2_: C, 70.92; H, 6.45; N, 6.89; found: C 71.06, H 6.24, N 6.96.

Ethyl 2-(2-methyl-1*H*-indol-1-yl) acetate **(15a)**

To a 57% NaH suspension in anhydrous DMF (0.72 g, 0.017 mol), according to the same procedure described for **14a**, 2-methyl-1*H*-indole **10** (1.5 g, 0.011 mol) and ethyl-2-bromoacetate **13a** (1.9 mL, 0.017 mol) were added. The title compound **15a** (836 mg, 35%) was obtained after purification by silica gel column chromatography using CHCl_3_/acetone 95:5 as eluent mixture; consistency: off-white oil; *R*_f_ = 0.61 (petroleum ether/EtOAc 9:1). ^1^H NMR (500 MHz, CDCl_3_) = δ: 1.26 (t, *J* = 7.1 Hz, 3H), 2.43 (s, 3H), 4.20 (q, *J* = 7.1 Hz, 2H), 4.64 (s, 2H), 6.31 (s, 1H), 7.08 (t, *J* = 7.5 Hz, 1H), 7.15 (t, *J* = 7.5 Hz, 1H), 7.20 (d, *J* = 8.1 Hz, 1H), 7.52 (d, *J* = 7.7 Hz, 1H) ppm; ^13^C NMR (125 MHz, CDCl_3_) = δ: 12.19, 13.79, 44.15, 61.05, 100.88, 108.12, 119.66, 119.70, 120.81, 128.02, 136.26, 136.90, 168.01. Elemental analysis: calcd for C_13_H_15_NO_2_: C, 71.87; H, 6.96; N, 6.45; found: C 71.64, H 6.78, N 6.54.

Ethyl 2-(3-methyl-1*H*-indol-1-yl) acetate **(16a)**

To a 57% NaH suspension in anhydrous DMF (1.44 g, 0.034 mol), according to the same procedure described for **14a**, 3-methyl-1*H*-indole **11** (3 g, 0.023 mol) and ethyl-2-bromoacetate **13a** (4.26 mL, 0.034 mol) were added. The title compound **16a** (1.15 g, 23%) was obtained after purification by silica gel column chromatography using petroleum ether/EtOAc 9:1 as eluent mixture; consistency: light yellow oil; *R*_f_ = 0.62 (petroleum ether/EtOAc 9:1). ^1^H NMR (500 MHz, CDCl_3_) = δ: 1.34 (t, *J* = 7.1 Hz, 3H), 2.44 (s, 3H), 4.27 (q, *J* = 7.1 Hz, 2H), 4.80 (s, 2H), 6.91 (s, 1H), 7.22–7.26 (m, 1H), 7.28–7.34 (m, 2H), 7.68–7.70 (m, 1H) ppm; ^13^C NMR (125 MHz, CDCl_3_) = δ: 9.42, 13.96, 47.36, 61.32, 108.61, 111.30, 118.98, 121.75, 125.87, 128.83, 136.67, 137.01, 168.66. Elemental analysis: calcd for C_13_H_15_NO_2_: C, 71.87; H, 6.96; N, 6.45; found: C 71.77, H 6.86, N 6.47.

Ethyl 2-(5-fluoro-1*H*-indol-1-yl) acetate **(17a)**

To a 57% NaH suspension in anhydrous DMF (0.701 g, 0.017 mol), according to the same procedure described for **14a**, 5-fluoro-1*H*-indole **12** (1.5 g, 0.011 mol) and ethyl-2-bromoacetate **13a** (1.85 mL, 0.017 mol) were added. The title compound **17a** (0.565 g, 23%) was obtained after purification by silica gel column chromatography using petroleum ether/acetone/EtOAc 8:1.5:0.5 as eluent mixture; consistency: yellow oil; *R*_f_ = 0.63 (petroleum ether/acetone 8:2). ^1^H NMR (500 MHz, CDCl_3_) = δ: 1.28 (t, *J* = 7.1 Hz, 3H), 4.23 (q, *J* = 7.1 Hz, 2H), 4.76 (s, 2H), 6.55 (d, *J* = 3.2 Hz, 1H), 7.02 (t, *J* = 9.2 Hz, 1H), 7.12 (d, *J* = 3.2 Hz, 1H), 7.18 (dd, *J* = 8.9 and 4.2 Hz, 1H), 7.35 (d, *J* = 9.2 Hz, 1H) ppm; ^13^C NMR (125 MHz, CDCl_3_) = δ: 13.88, 47.67, 61.46, 102.09 (d, *J* = 4.6 Hz), 105.62 (d, *J* = 23.4 Hz), 109.63 (d, *J* = 9.9 Hz), 110.02 (d, *J* = 26.4 Hz), 128.82, 130.22, 133.04, 157.91 (d, *J* = 234.0 Hz), 168.26. Elemental analysis: calcd for C_12_H_12_FNO_2_: C, 65.15; H, 5.47; N, 8.59; found: C 64.92, H 5.58, N 8.43.

Ethyl 3-(1*H*-indol-1-yl) propanoate **(14b)**

To a 57% NaH suspension in anhydrous DMF (1.05 g, 0.026 mol), according to the same procedure described for **14a**, 1*H*-indole **9** and ethyl-3-bromopropionate **13b** (3.27 mL, 0.026 mol) were added. The title compound **14b** (1.22 g, 33%) was obtained after purification by silica gel column chromatography using petroleum ether/EtOAc/acetone 9:0.5:0.5 as eluent mixture; consistency: light yellow oil; *R*_f_ = 0.7 (petroleum ether/EtOAc 9:1). ^1^H NMR (500 MHz, CDCl_3_) = δ: 1.20 (t, *J* = 7.1 Hz, 3H), 2.82 (t, *J* = 6.9 Hz, 2H), 4.11 (q, *J* = 7.1 Hz, 2H), 4.46 (t, *J* = 6.9 Hz, 2H), 6.48 (s, 1H), 7.09–7.14 (m, 2H), 7.22 (t, *J* = 7.5 Hz, 1H), 7.35 (d, *J* = 8.2 Hz, 1H), 7.62 (d, *J* = 7.9Hz, 1H); ppm; ^13^C NMR (125 MHz, CDCl_3_) = δ: 14.17, 25.95, 29.69, 41.82, 60.98, 101.55, 109.07, 119.46, 121.03, 121.59, 122.42, 127.89, 136.58, 170.66 ppm. Elemental analysis: calcd for C_13_H_15_NO_2_: C, 71.87; H, 6.96; N, 6.45; found: C 71.78, H 6.86, N 6.54.

Ethyl 3-(2-methyl-1*H*-indol-1-yl) propanoate **(15b)**

To a 57% NaH suspension in anhydrous DMF (0.96 g, 0.023 mol), according to the same procedure described for **14a**, 2-methyl-1*H*-indole **10** (2 g, 0.015 mol) and ethyl-3-bromopropionate **13b** (2.93 mL, 0.023 mol) were added. The title compound **15b** (971 mg, 28%) was obtained after purification by silica gel column chromatography using petroleum ether/EtOAc 65:35 as eluent mixture; consistency: yellow oil; *R*_f_ = 0.51 (petroleum ether/EtOAc 65:35). ^1^H NMR (500 MHz, CDCl_3_) = δ: 1.23 (t, *J* = 7.1 Hz, 3H), 2.38 (s, 3H), 2.68 (t, *J* = 7.3 Hz, 2H), 4.13 (q, *J* = 7.1 Hz, 2H), 4.36 (t, *J* = 7.5 Hz, 2H), 6.87 (s, 1H), 7.00 (t, *J* = 7.1 Hz, 1H), 7.13 (t, *J* = 7.1 Hz, 1H), 7.27 (d, *J* = 8.2 Hz, 1H), 7.58 (d, *J* = 8.2 Hz, 1H) ppm; ^13^C NMR (125 MHz, CDCl_3_) = δ: 11.85, 14.09, 34.53, 38.46, 60.93, 109.06, 112.41, 119.05, 119.76, 119.83, 121.04, 125.30, 137.38, 170.88. Elemental analysis: calcd for C_14_H_17_NO_2_: C, 72.70; H, 7.41; N, 6.06; found: C 72.69, H 7.36, N 5.89.

Ethyl 3-(3-methyl-1*H*-indol-1-yl) propanoate **(16b)**

To a 57% NaH suspension in anhydrous DMF (0.96 g, 0.023 mol), according to the same procedure described for **14a**, 3-methyl-1*H*-indole **11** (2 g, 0.015 mol) and ethyl-3-bromopropionate **13b** (2.93 mL, 0.023 mol) were added. The title compound **16b** (1.25 g, 36%) was obtained after purification by silica gel column chromatography using petroleum ether/EtOAc 9:1 as eluent mixture; consistency: light yellow oil; *R*_f_ = 0.62 (petroleum ether/EtOAc 9:1). ^1^H NMR (500 MHz, CDCl_3_) = δ: 1.31 (t, *J* = 6.9 Hz, 3H), 2.44 (s, 3H), 2.85 (t, *J* = 6.9 Hz, 2H), 4.21 (q, *J* = 13 Hz, 2H), 4.45 (t, *J* = 6.9 Hz, 2H), 6.98 (s, 1H), 7.24 (t, *J* = 7.5 Hz, 1H), 7.33 (t, *J* = 7.5 Hz, 1H), 7.40 (d, *J* = 8.0 Hz, 1H), 7.69 (d, *J* = 8.0 Hz, 1H); ^13^C NMR (125 MHz, CDCl_3_) = δ: 9.65, 14.10, 35.03, 41.70, 60.97, 109.05, 110.64, 118.87, 119.18, 121.70, 125.60, 128.89, 136.23, 170.99. Elemental analysis: calcd for C_14_H_17_NO_2_: C, 72.70; H, 7.41; N, 6.06; found: C 72.84, H 7.12, N 6.09.

Ethyl 3-(5-fluoro-1*H*-indol-1-yl) propanoate **(17b)**

To a 57% NaH suspension in anhydrous DMF (0.467 g, 0.011 mol), according to the same procedure described for **14a**, 5-fluoro-1*H*-indole **12** (1 g, 0.007 mol) and ethyl-3-bromopropionate **13b** (1.42 mL, 0.011 mol) were added. The title compound **17b** (0.56 g, 32%) was obtained after purification by silica gel column chromatography using petroleum ether/acetone/EtOAc 9:0.5:0.5 as eluent mixture; consistency: pale yellow oil; *R*_f_ = 0.63 (petroleum ether/acetone 8:2). ^1^H NMR (500 MHz, CDCl_3_) = δ: 1.21 (t, *J* = 7.2 Hz, 3H), 2.80 (t, *J* = 6.8 Hz, 2H), 4.12 (q, *J* = 7.2 Hz, 2H), 4.43 (t, *J* = 6.8 Hz, 2H), 6.44 (d, *J* = 3.1 Hz, 1H), 6.97 (t, *J* = 6.7 Hz, 1H), 7.17 (d, *J* = 3.1 Hz, 1H), 7.24–7.28 (m, 1H) ppm; ^13^C NMR (125 MHz, CDCl_3_) = δ: 14.06, 35.02, 42.05, 60.92, 101.53 (d, *J* = 4.7 Hz), 105.72 (d, *J* = 23.3 Hz), 109.69 (d, *J* = 9.8 Hz), 109.94, 128.88, 129.47, 132.32, 157.87 (d, *J* = 234.1 Hz), 171.08. Elemental analysis: calcd for C_13_H_14_FNO_2_: C, 66.37; H, 6.00; N, 5.95; found: C 66.58, H 6.12, N 5.91.

2-(1*H*-Indol-1-yl) acetic acid **(18a)**

To a solution of **14a** (1.74 g, 0.0086 mol) in a mixture methanol/water/dioxane (1:1:1), placed in an ice bath at 0 °C, LiOH as powder (1.03 g, 0.043 mol) was added. The reaction mixture was allowed to stir overnight at room temperature. After overnight stirring, solvents were removed in vacuo and the residue was extracted in Et_2_O and water. The aqueous phase was brought to pH = 1–2 by addition of a 10% KHSO_4_ solution and extracted with EtOAc. The organic phase was then dried over Na_2_SO_4_, filtered and evaporated in vacuo to obtain the pure carboxylic acid **18a** (827 mg, 55%); consistency: yellow powder. ^1^H NMR (500 MHz, DMSO) = δ: 5.01 (s, 2H), 6.44 (s, 1H), 7.03 (t, *J* = 7.0 Hz, 1H), 7.12 (t, *J* = 6.6 Hz, 1H), 7.32–7.34 (m, 1H), 7.37 (d, *J* = 8.2 Hz, 1H), 7.55 (d, *J* = 6.6 Hz, 1H) ppm. Elemental analysis: calcd for C_10_H_9_NO_2_: C, 68.56; H, 5.18; N, 8.00; found: C 68.32, H 5.12, N 8.13.

2-(2-Methyl-1*H*-indol-1-yl) acetic acid **(19a)**

To a solution of **15a** (238 mg, 1.1 mmol), according to the same procedure described for **18a**, LiOH as powder (131.17 mg, 5.48 mmol) was added and the pure carboxylic acid **19a** (195 mg, 94%) was obtained; consistency: burgundy powder. ^1^H NMR (500 MHz, CDCl_3_) = δ: 2.40 (s, 3H), 4.83 (s, 2H), 6.31 (s, 1H), 7.07–7.11 (m, 1H), 7.13–7.19 (m, 2H), 7.53 (d, *J* = 7.7 Hz, 1H) ppm. Elemental analysis: calcd for C_11_H_11_NO_2_: C, 68.83; H, 5.86; N, 7.40; found: C 68.66, H 5.93, N 7.46.

2-(3-Methyl-1*H*-indol-1-yl) acetic acid **(20a)**

To a solution of **16a** (1.15 g, 0.005 mol), according to the same procedure described for **18a**, LiOH as powder (634 mg, 0.026 mol) was added and the pure carboxylic acid **20a** (195 mg, 94%) was obtained; consistency: yellow powder. ^1^H NMR (500 MHz, CDCl_3_) = δ: 2.32 (s, 3H), 4.83 (s, 2H), 6.84 (s, 1H), 7.12–7.15 (m, 1H), 7.18–7.24 (m, 2H), 7.57 (d, *J* = 7.7 Hz, 1H) ppm. Elemental analysis: calcd for C_11_H_11_NO_2_: C, 68.83; H, 5.86; N, 7.40; found: C 68.78, H 6.02, N 7.47.

2-(5-Fluoro-1*H*-indol-1-yl) acetic acid **(21a)**

To a solution of **17a** (565 mg, 2.55 mmol), according to the same procedure described for **18a**, LiOH as powder (305.83 mg, 12.79 mmol) was added and the pure carboxylic acid **21a** (332 mg, 67%) was obtained; consistency: orange powder. ^1^H NMR (500 MHz, CDCl_3_) = δ: 4.86 (s, 2H), 6.53 (d, *J* = 3.2 Hz, 1H), 6.98 (t, *J* = 9.2 Hz, 1H), 7.09 (d, *J* = 3.2 Hz, 1H), 7.14 (dd, *J* = 8.9 and 4.2 Hz, 1H), 7.28 (dd, *J* = 9.2 and 2.3 Hz, 1H) ppm. Elemental analysis: calcd for C_10_H_8_FNO_2_: C, 62.18; H, 4.17; N, 9.83; found: C 61.99, H 4.23, N 9.86.

3-(1*H*-Indol-1-yl) propanoic acid **(18b)**

To a solution of **14b** (538 mg, 2.48 mmol), according to the same procedure described for **18a**, LiOH as powder (296.54 mg, 12.38 mmol) was added and the pure carboxylic acid **18b** (205 mg, 44%) was obtained; consistency: orange powder. ^1^H NMR (500 MHz, CDCl_3_) = δ: 2.88 (t, *J* = 6.9 Hz, 2H), 4.45 (t, *J* = 6.9 Hz, 2H), 6.53 (s, 1H), 7.14–7.18 (m, 2H), 7.26 (t, *J* = 7.1 Hz, 1H), 7.38 (d, *J* = 8.2 Hz, 1H), 7.68 (d, *J* = 7.1 Hz, 1H) ppm. Elemental analysis: calcd for C_11_H_11_NO_2_: C, 69.83; H, 5.86; N, 7.40; found: C 69.78, H 6.00, N 7.42.

3-(2-Methyl-1*H*-indol-1-yl) propanoic acid **(19b)**

To a solution of **15b** (121 mg, 0.52 mmol), according to the same procedure described for **18a**, LiOH as powder (62.65 mg, 2.62 mmol) was added and the pure carboxylic acid **19b** (105 mg, 97%) was obtained; consistency: orange powder. ^1^H NMR (500 MHz, CDCl_3_) = d: 2.33 (s, 3H), 2.68 (t, *J* = 7.4 Hz, 2H), 4.31 (t, *J* = 7.4 Hz, 2H), 6.84–6.87 (m, 1H), 6.99 (t, *J* = 7.6 Hz, 1H), 7.11 (t, *J* = 7.6 Hz, 1H), 7.25 (d, *J* = 7.1 Hz, 1H), 7.48 (d, *J* = 7.1 Hz, 1H) ppm. Elemental analysis: calcd for C_12_H_13_NO_2_: C, 70.92; H, 6.45; N, 6.89; found: C 70.82, H 6.36, N 7.02.

3-(3-Methyl-1*H*-indol-1-yl) propanoic acid **(20b)**

To a solution of **16b** (824 mg, 3.56 mmol), according to the same procedure described for **18a**, LiOH as powder (426.62 mg, 17.81 mmol) was added and the pure carboxylic acid **20b** (704 mg, 97%) was obtained; consistency: light yellow powder. ^1^H NMR (500 MHz, CDCl_3_) = d: 2.33 (s, 3H), 2.86 (t, *J* = 6.8 Hz, 2H), 4.40 (t, *J* = 6.9 Hz, 2H), 6.90 (s, 1H), 7.14 (t, *J* = 7.0 Hz, 1H), 7.24 (t, *J* = 7.6 Hz, 1H), 7.32 (d, *J* = 8.2 Hz, 1H), 7.59 (d, *J* = 7.9 Hz, 1H) ppm. Elemental analysis: calcd for C_12_H_13_NO_2_: C, 70.92; H, 6.45; N, 6.89; found: C 70.99, H 6.56, N 7.00.

3-(5-Fluoro-1*H*-indol-1-yl) propanoic acid **(21b)**

To a solution of **17b** (560 mg, 2.38 mmol), according to the same procedure described for **18a**, LiOH as powder (285.08 mg, 11.9 mmol) was added and the pure carboxylic acid **21b** (327 mg, 66%) was obtained; consistency: yellow powder. ^1^H NMR (500 MHz, CDCl_3_) = δ: 2.87 (t, *J* = 6.8 Hz, 2H), 4.43 (t, *J* = 6.8 Hz, 2H), 6.44 (d, *J* = 3.1 Hz, 1H), 6.96 (t, *J* = 6.7 Hz, 1H), 7.16 (d, *J* = 3.1 Hz, 1H), 7.23–7.28 (m, 2H) ppm. Elemental analysis: calcd for C_11_H_10_FNO_2_: C, 63.76; H, 4.86; N, 9.17; found: C 63.52, H 5.01, N 9.13.

*N*-((*S*)-1-Cyano-3-phenylpropyl)2-(1*H*-indol-1-yl)acetamide **(1a)**

To a solution of acid 2-(1*H*-indol-1-il)acetic **18a** (32.8 mg, 0.19 mmol) in anhydrous DCM/DMF, placed in an ice bath at 0 °C, HOBt (31.62 mg, 0.23 mmol) and EDCI (44.87 mg, 0.23 mmol) were added. After 10 min, (*S*)-2-amino-4-phenylbutanitrile **22** (25 mg, 0.16 mmol) and DIPEA (33.28 µL, 0.19 mmol) were added and the reaction mixture was stirred overnight at room temperature. After overnight stirring, the solvents were evaporated in vacuo; then the residue was diluted with EtOAc, washed with brine, dried over Na_2_SO_4_, filtered, and concentrated in vacuo. The crude residue was purified by silica gel column chromatography using petroleum ether/EtOAc 7:3 to obtain the pure coupling product **1a** (31 mg, 61%); consistency: yellow powder; M.p.: 120–121 °C, R_f_ = 0.52 (petroleum ether/EtOAc 7:3); ^1^H NMR (500 MHz, CDCl_3_) = δ: 1.84–1.91 (m, 1H), 1.91–1.99 (m, 1H), 2.49–2.60 (m, 2H), 4.73–4.79 (m, 1H), 4.82 (s, 2H), 5.50 (bs, 1H), 6.66 (s, 1H), 7.00 (d, *J* = 7.5 Hz, 2H), 7.02–7.05 (m, 1H), 7.18–7.24 (m, 4H), 7.28–7.32 (m, 2H), 7.69 (d, *J* = 7.5 Hz, 1H) ppm; ^13^C NMR (125 MHz, CDCl_3_) = δ: 31.36, 34.29, 39.95, 49.83, 104.25, 109.05, 117.72, 120.99, 121.70, 123.23, 126.84, 128.19, 128.35, 128.86, 128.99, 136.21, 138.82, 168.10. Elemental analysis: calcd for C_20_H_19_N_3_O: C, 75.59; H, 6.03; N, 13.24; found: C 75.42, H 6.12, N 13.42.

*N*-((*S*)-1-Cyano-3-phenylpropyl)-2-(2-methyl-1*H*-indol-1-yl)acetamide **(2a)**

According to the same procedure described for **1a**, a solution of 2-(2-methyl-1*H*-indol-1-yl)acetic acid **19a** (35.43 mg, 0.19 mmol) was reacted with (*S*)-2-amino-4-phenylbutanitrile **22** (25 mg, 0.16 mmol) and a crude residue was obtained. It was purified by silica gel column chromatography using petroleum ether/EtOAc 8:2 to obtain the pure coupling product **2a** (17 mg, 32%); consistency: yellow powder; M.p.: 131–133 °C, *R*_f_ = 0.37 (petroleum ether/EtOAc 8:2). ^1^H NMR (500 MHz, CDCl_3_) = δ: 1.82–1.90 (m, 1H), 1.90–1.98 (m, 1H), 2.39 (s, 3H), 2.48–2.59 (m, 2H), 4.72–4.78 (m, 3H), 5.46 (bs, 1H), 6.39 (s, 1H), 7.00 (d, *J* = 7.6 Hz, 2H), 7.13–7.18 (m, 2H), 7.18–7.22 (m, 2H), 7.24–7.27 (m, 2H), 7.58 (d, *J* = 7.6 Hz, 1H) ppm; ^13^C NMR (125 MHz, CDCl_3_) = δ: 12.69, 31.39, 34.46, 39.94, 46.77, 102.78, 108.61, 117.71, 120.61, 121.11, 122.29, 126.92, 128.38, 128.91, 136.11, 136.87, 138.77, 168.30. Elemental analysis: calcd for C_21_H_21_N_3_O: C, 76.11; H, 6.39; N, 12.68; found: C 76.03, H 6.19, N 12.87.

*N*-((*S*)-1-Cyano-3-phenylpropyl)-2-(3-methyl-1*H*-indol-1-yl)acetamide **(3a)**

According to the same procedure described for **1a**, a solution of 2-(3-methyl-1*H*-indol-1-yl)acetic acid **20a** (35.43 mg, 0.19 mmol) was reacted with (*S*)-2-amino-4-phenylbutanitrile **22** (25 mg, 0.16 mmol) and a crude residue was obtained. It was purified by silica gel column chromatography using petroleum ether/EtOAc 8:2 to obtain the pure coupling product **3a** (19 mg, 36%); consistency: pale yellow powder; M.p.: 114–115 °C, *R*_f_ = 0.35 (petroleum ether/EtOAc 8:2). ^1^H NMR (500 MHz, CDCl_3_) = δ: 1.84–1.91 (m, 1H), 1.92–1.99 (m, 1H), 2.35 (s, 3H), 2.51–2.59 (m, 2H), 4.74–4.79 (m, 3H), 5.55 (bs, 1H), 6.79 (s, 1H), 7.00 (d, *J* = 7.7 Hz, 2H), 7.20 (d, *J* = 7.8 Hz, 2H), 7.23–7.29 (m, 4H), 7.63 (d, *J* = 7.7 Hz, 1H) ppm; ^13^C NMR (125 MHz, CDCl_3_) = δ: 9.59, 31.28, 34.25, 39.80, 49.51, 108.79, 113.43, 117.65, 119.67, 120.22, 123.06, 125.50, 126.72, 128.24, 128.74, 129.34, 136.50, 138.75, 168.42. Elemental analysis: calcd for C_21_H_21_N_3_O: C, 76.11; H, 6.39; N, 12.68; found: C 76.24, H 6.42, N 12.85.

*N*-((*S*)-1-Cyano-3-phenylpropyl)-2-(5-fluoro-1*H*-indol-1-yl)acetamide **(4a)**

According to the same procedure described for **1a**, a solution of 2-(5-fluoro-1*H*-indol-1-yl) acetic acid **21a** (57.87 mg, 0.30 mmol) was reacted with (*S*)-2-amino-4-phenylbutanitrile **22** (40 mg, 0.25 mmol) and a crude residue was obtained. It was purified by silica gel column chromatography using petroleum ether/EtOAc 7:3 to obtain the pure coupling product **4a** (40.3 mg, 32%); consistency: white powder; M.p.: 150–152 °C, *R*_f_ = 0.42 (petroleum ether/EtOAc 7:3). ^1^H NMR (500 MHz, CDCl_3_) = δ: 1.85–1.93 (m, 1H), 1.93–2.00 (m, 1H), 2.57 (t, *J* = 7.6 Hz, 2H), 4.73–4.77 (m, 1H), 4.78 (s, 2H), 5.53 (bs, 1H), 6.60 (s, 1H), 7.02 (d, *J* = 7.4 Hz, 2H), 7.04–7.08 (m, 1H), 7.13–7.28 (m, 5H), 7.33 (d, *J* = 9.2 Hz, 1H) ppm; ^13^C NMR (125 MHz, CDCl_3_) = δ: 31.46, 34.38, 40.06, 50.13, 104.30 (d, *J* = 4.5 Hz), 106.74 (d, *J* = 23.6 Hz), 109.83 (d, *J* = 9.9 Hz), 111.77 (d, *J* = 26.6 Hz), 117.65, 126.97, 128.36, 128.97, 129.82, 131.03, 132.81, 138.76, 158.64 (d, *J* = 236.7 Hz), 167.74. Elemental analysis: calcd for C_20_H_18_FN_3_O: C, 71.63; H, 5.41; N, 12.53; found: C 71.42, H 5.26, N 12.63.

*N*-((*S*)-1-Cyano-3-phenylpropyl)-3-(1*H*-indol-1-yl)propanamide **(1b)**

According to the same procedure described for **1a**, a solution of 3-(1*H*-indol-1-yl)propanoic acid **18b** (42.51 mg, 0.22 mmol) was reacted with (*S*)-2-amino-4-phenylbutanitrile **22** (30 mg, 0.19 mmol) and a crude residue was obtained. It was purified by silica gel column chromatography using petroleum ether/EtOAc 7:3 to obtain the pure coupling product **1b** (26.4 mg, 21%); consistency: yellow powder; M.p.: 155–156 °C, *R*_f_ = 0.39 (petroleum ether/EtOAc 7:3). ^1^H NMR (500 MHz, CDCl_3_) = δ: 1.75–1.89 (m, 2H), 2.52–2.63 (m, 4H), 4.39–4.45 (m, 1H), 4.48–4.54 (m, 1H), 4.70 (dd, *J* = 14.9 Hz e 7.5 Hz, 1H), 5.35 (bs, 1H), 6.48 (d, *J* = 3.1 Hz, 1H), 7.04–7.07 (m, 3H), 7.11 (t, *J* = 7.0 Hz, 1H), 7.18–7.25 (m, 4H), 7.31 (d, *J* = 8.0 Hz, 1H), 7.61 (d, *J* = 8.0 Hz, 1H) ppm; ^13^C NMR (125 MHz, CDCl_3_) = δ: 31.51, 34.04, 37.11, 40.27, 42.38, 102.10, 109.21, 119.89, 121.42, 121.98, 126.88, 128.13, 128.48, 128.93, 135.58, 139.14, 169.84. Elemental analysis: calcd for C_21_H_21_N_3_O: C, 76.11; H, 6.39; N, 12.68; found: C 76.13, H 6.29, N 12.86.

*N*-((*S*)-1-Cyano-3-phenylpropyl)-3-(2-methyl-1*H*-indol-1-yl)propanamide **(2b)**

According to the same procedure described for **1a**, a solution of 3-(2-methyl-1*H*-indol-1-yl)propanoic acid **19b** (53.28 mg, 0.26 mmol) was reacted with (*S*)-2-amino-4-phenylbutanitrile **22** (35 mg, 0.22 mmol) and a crude residue was obtained. It was purified by silica gel column chromatography using petroleum ether/EtOAc 5:5 to obtain the pure coupling product **2b** (32 mg, 42%); consistency: yellow powder; M.p.: 144–145 °C, *R*_f_ = 0.44 (petroleum ether/EtOAc 5:5). ^1^H NMR (500 MHz, CDCl_3_) = δ: 1.80–1.87 (m, 2H), 2.31 (s, 3H), 2.37–2.49 (m, 2H), 2.48–2.64 (m, 2H), 4.25–4.33 (m, 1H), 4.34–4.43 (m, 1H), 4.61 (dd, *J* = 15.2 Hz e 7.5 Hz, 1H), 5.86 (s, 1H), 6.86 (t, *J* = 8.0 Hz, 1H), 6.95 (t, *J* = 8.0 Hz, 1H), 7.04–7.11 (m, 3H), 7.17–7.23 (m, 3H), 7.45 (d, *J* = 8.0 Hz, 1H), 7.67 (d, *J* = 7.8 Hz, 1H) ppm; ^13^C NMR (125 MHz, CDCl_3_) = δ: 11.93, 31.54, 33.81, 36.46, 39.05, 40.23, 109.38, 112.58, 118.26, 119.16, 120.12, 121.31, 125.41, 126.79, 128.52, 128.86, 135.70, 137.84, 139.35, 170.04. Elemental analysis: calcd for C_22_H_23_N_3_O: C, 76.49; H, 6.71; N, 12.16; found: C 76.26, H 6.84, N 12.22.

*N*-((*S*)-1-Cyano-3-phenylpropyl)-3-(3-methyl-1*H*-indol-1-yl)propanamide **(3b)**

According to the same procedure described for **1a**, a solution of 3-(3-methyl-1*H*-indol-1-yl)propanoic acid **20b** (53.28 mg, 0.26 mmol) was reacted with (*S*)-2-amino-4-phenylbutanitrile **22** (35 mg, 0.22 mmol) and a crude residue was obtained. It was purified by silica gel column chromatography using petroleum ether/EtOAc 65:35 to obtain the pure coupling product **3b** (28.8 mg, 38%); consistency: light yellow powder; M.p.: 130–132 °C, *R*_f_ = 0.48 (petroleum ether/EtOAc 65:35). ^1^H NMR (500 MHz, CDCl_3_) = δ: 1.75–1.86 (m, 2H), 2.26 (s, 3H), 2.49–2.55 (m, 2H), 2.55–2.61 (m, 2H), 4.31–4.38 (m, 1H), 4.39–4.47 (m, 1H), 4.68 (dd, *J* = 14.8 Hz and 7.6 Hz, 1H), 5.40 (bs, 1H), 6.83 (s, 1H), 7.05 (d, *J* = 7.4 Hz, 2H), 7.10 (t, *J* = 7.4 Hz, 1H), 7.20 (t, *J* = 7.2 Hz, 2H), 7.23–7.27 (m, 3H), 7.54 (d, *J* = 7.9 Hz, 1H) ppm; ^13^C NMR (125 MHz, CDCl_3_) = δ: 9.49, 31.34, 33.95, 37.02, 40.05, 42.01, 108.89, 111.09, 117.85, 119.01, 119.32, 121.75, 125.55, 126.70, 128.30, 128.75, 129.00, 135.72, 139.01, 169.91. Elemental analysis: calcd for C_22_H_23_N_3_O: C, 76.49; H, 6.71; N, 12.16; found: C 76.58, H 6.74, N 12.02.

*N*-((*S*)-1-Cyano-3-phenylpropyl)-3-(5-fluoro-1*H*-indol-1-yl)propanamide **(4b)**

According to the same procedure described for **1a**, a solution of 3-(5-fluoro-1*H*-indol-1-yl) propanoic acid **21b** (62.07 mg, 0.30 mmol) was reacted with (*S*)-2-amino-4-phenylbutanitrile **22** (40 mg, 0.25 mmol) and a crude residue was obtained. It was purified by silica gel column chromatography using petroleum ether/EtOAc 6:4 to obtain the pure coupling product **4b** (33 mg, 39%); consistency: pale yellow powder; M.p.: 142–143 °C, *R*_f_ = 0.46 (petroleum ether/EtOAc 6:4). ^1^H NMR (500 MHz, CDCl_3_) = δ: 1.78–1.85 (m, 1H), 1.85–1.92 (m, 1H), 2.52 (m, 2H), 2.61 (m, 2H), 4.34–4.41 (m, 1H), 4.43–4.50 (m, 1H), 4.68 (dd, *J* = 14.3 and 7.1 Hz, 1H), 5.66 (bs, 1H), 6.42 (s, 1H), 6.95 (t, *J* = 9.0 Hz, 1H), 7.05 (d, *J* = 7.3 Hz, 2H), 7.10 (s, 1H), 7.28–7.19 (m, 5H) ppm; ^13^C NMR (125 MHz, CDCl_3_) = δ: 31.48, 34.05, 36.95, 40.25, 42.50, 101.97 (d, *J* = 4.8 Hz), 105.95, 106.13, 109.87 (d, *J* = 9.8 Hz), 110.29 (d, *J* = 26.3 Hz), 118.03, 126.88, 128.42, 128.93, 129.70, 132.26, 139.07, 158.01 (d, *J* = 234.5 Hz), 169.83. Elemental analysis: calcd for C_21_H_20_FN_3_O: C, 72.19; H, 5.77; N, 12.03; found: C 72.37, H 5.45, N 12.02.

*N*-(1-Cyanocyclopropyl)-2-(1*H*-indol-1-yl)acetamide **(5a)**

According to the same procedure described for **1a**, a solution of 2-(1*H*-indol-1-yl)acetic acid **18a** (25 mg, 0.14 mmol) was reacted with 1-aminocyclopropanocarbonitrile **23** (20.3 mg, 0.17 mmol) and a crude residue was obtained. It was purified by silica gel column chromatography using petroleum ether/EtOAc 5:5 to obtain the pure coupling product **5a** (18 mg, 54%); consistency: light yellow powder; M.p.: 179–181 °C, *R*_f_ = 0.39 (petroleum ether/EtOAc 5:5). ^1^H NMR (500 MHz, CDCl_3_) = δ: 1.06 (t, *J* = 5.8 Hz, 2H), 1.48 (t, *J* = 5.8 Hz, 2H), 4.81 (s, 2H), 5.84 (bs, 1H), 6.63 (s, 1H), 7.03–7.06 (m, 1H), 7.18–7.23 (m, 2H), 7.28 (t, *J* = 7.5 Hz, 1H), 7.68 (d, *J* = 8.0 Hz, 1H) ppm; ^13^C NMR (125 MHz, CDCl_3_) = δ: 16.83, 20.23, 49.92, 104.13, 108.84, 119.23, 120.85, 121.55, 123.16, 128.86, 127.98, 136.00, 169.18. Elemental analysis: calcd for C_14_H_13_N_3_O: C, 70.28; H, 5.48; N, 17.56; found: C 70.36, H 5.69, N 17.65.

*N*-(1-Cyanocyclopropyl)-2-(2-methyl-1*H*-indol-1-yl)acetamide **(6a)**

According to the same procedure described for **1a**, a solution of 2-(2-methyl-1*H*-indol-1-yl)acetic acid **19a** (40 mg, 0.21 mmol) was reacted with 1-aminocyclopropanocarbonitrile **23** (30.08 mg, 0.25 mmol) and a crude residue was obtained. It was purified by silica gel column chromatography using petroleum ether/EtOAc 5:5 to obtain the pure coupling product **6a** (36 mg, 68%); consistency: light orange powder; M.p.: 178–180 °C, *R*_f_ = 0.45 (petroleum ether/EtOAc 5:5). ^1^H NMR (500 MHz, CDCl_3_) = δ: 1.16 (t, *J* = 6.0 Hz, 2H), 1.49 (t, *J* = 6.0 Hz, 2H), 2.56 (s, 3H), 4.76 (s, 2H), 6.22 (s, 1H), 6.98 (t, *J* = 7.7 Hz, 1H), 7.04 (t, *J* = 7.7 Hz, 1H), 7.27 (d, *J* = 7.7 Hz, 1H), 7.42 (d, *J* = 7.7 Hz, 1H) ppm; ^13^C NMR (125 MHz, CDCl_3_) = δ: 12.46, 15.78, 19.92, 45.35, 100.04, 109.12, 119.28, 120.37, 120.74, 127.83, 137.32, 137.37, 169.37. Elemental analysis: calcd for C_15_H_15_N_3_O: C, 71.13; H, 5.97; N, 16.59; found: C 70.99, H 5.98, N 16.88.

*N*-(1-Cyanocyclopropyl)-2-(3-methyl-1*H*-indol-1-yl)acetamide **(7a)**

According to the same procedure described for **1a**, a solution of 2-(3-methyl-1*H*-indol-1-yl)acetic acid **20a** (50 mg, 0.26 mmol) was reacted with 1-aminocyclopropanocarbonitrile **23** (37.6 mg, 0.32 mmol) and a crude residue was obtained. It was purified by silica gel column chromatography using petroleum ether/EtOAc 5:5 to obtain the pure coupling product **7a** (23.7 mg, 36%); consistency: off-white powder; M.p.: 193–195 °C, *R*_f_ = 0.52 (EtOAc/petroleum ether 6:4). ^1^H NMR (500 MHz, CDCl_3_) = δ: 1.06 (t, *J* = 6.2 Hz, 2H), 1.47 (t, *J* = 6.2 Hz, 2H), 2.34 (s, 3H), 4.73 (s, 2H), 5.89 (bs, 1H), 6.80 (s, 1H), 7.14–7.21 (m, 2H), 7.25–7.29 (m, 1H), 7.61 (d, *J* = 7.9 Hz, 1H) ppm; ^13^C NMR (125 MHz, CDCl_3_) = δ: 9.63, 16.85, 20.24, 49.60, 108.54, 108.65, 113.45, 119.04, 119.31, 120.10, 122.81, 125.42, 129.31, 136.37, 169.45. Elemental analysis: calcd for C_15_H_15_N_3_O: C, 71.13; H, 5.97; N, 16.59; found: C 70.98, H 5.97, N 16.54.

*N*-(1-cyanocyclopropyl)-2-(5-fluoro-1*H*-indol-1-yl)acetamide **(8a)**

According to the same procedure described for **1a**, a solution of 2-(5-fluoro-1*H*-indol-1-yl) acetic acid **21a** (50 mg, 0.26 mmol) was reacted with 1-aminocyclopropanocarbonitrile **23** (36.83 mg, 0.31 mmol) and a crude residue was obtained. It was purified by silica gel column chromatography using EtOAc/petroleum ether 7:3 to obtain the pure coupling product **8a** (28.4 mg, 42%); consistency: white powder; M.p.: 180–181 °C, *R*_f_ = 0.59 (EtOAc/petroleum ether 7:3). ^1^H NMR (500 MHz, CDCl_3_) = δ: 1.07 (t, *J* = 6.7 Hz, 2H), 1.49 (t, *J* = 6.7 Hz, 2H), 4.79 (s, 2H), 5.82 (bs, 1H), 6.59 (s, 1H), 7.02 (t, *J* = 8.9 Hz, 1H), 7.09 (s, 1H), 7.13 (dd, *J* = 8.9 and 3.9 Hz, 1H), 7.31 (d, *J* = 8.9 Hz, 1H) ppm; ^13^C NMR (125 MHz, CDCl_3_) = δ: 16.93, 21.32, 49.94, 103.01 (d, *J* = 4.7 Hz), 106.34 (d, *J* = 23.7 Hz), 110.95 (d, *J* = 5.9 Hz), 111.06, 121.09, 130.68 (d, *J* = 10.2 Hz), 131.98, 134.75, 159.43 (d, *J* = 233.0 Hz), 171.81. Elemental analysis: calcd for C_14_H_12_FN_3_O: C, 65.36; H, 4.70; N, 16.33; found: C 65.22, H 4.55, N 16.51.

*N*-(1-Cyanocyclopropyl)-3-(1*H*-indol-1-yl)propanamide **(5b)**

According to the same procedure described for **1a**, a solution of 3-(1*H*-indol-1-yl)propanoic acid **18b** (50 mg, 0.26 mmol) was reacted with 1-aminocyclopropanocarbonitrile **23** (37.6 mg, 0.32 mmol) and a crude residue was obtained. It was purified by silica gel column chromatography using petroleum ether/EtOAc 5:5 to obtain the pure coupling product **5b** (29.9 mg, 45%); consistency: yellow oil; M.p.: 139–141 °C, *R*_f_ = 0.43 (EtOAc/petroleum ether 6:4). ^1^H NMR (500 MHz, CDCl_3_) = δ: 0.82 (t, *J* = 6.0 Hz, 2H), 1.32 (t, *J* = 6.0 Hz, 2H), 2.53 (t, *J* = 6.3 Hz, 2H), 4.42 (t, *J* = 6.3 Hz, 2H), 5.88 (bs, 1H), 6.46 (s, 1H), 7.05 (s, 1H), 7.10 (t, *J* = 7.9, Hz, 1H), 7.20 (t, *J* = 7.9 Hz, 1H), 7.29 (d, *J* = 7.9 Hz, 1H), 7.60 (d, *J* = 7.9 Hz, 1H) ppm; ^13^C NMR (125 MHz, CDCl_3_) = δ: 16.50, 20.18, 36.81, 42.14, 101.76, 109.17, 119.68, 119.74, 121.15, 121.79, 128.09, 128.64, 135.42, 171.28. Elemental analysis: calcd for C_15_H_15_N_3_O: C, 71.13; H, 5.97; N, 16.59; found: C 71.23, H 5.77, N 16.68.

*N*-(1-Cyanocyclopropyl)-3-(2-methyl-1*H*-indol-1-yl)propanamide **(6b)**

According to the same procedure described for **1a**, a solution of 3-(2-methyl-1*H*-indol-1-yl)propanoic acid **19b** (50 mg, 0.25 mmol) was reacted with 1-aminocyclopropanocarbonitrile **23** (35 mg, 0.30 mmol) and a crude residue was obtained. It was purified by silica gel column chromatography using EtOAc/petroleum ether 7:3 to obtain the pure coupling product **6b** (26 mg, 39%); consistency: yellow powder; M.p.: 124–126 °C, *R*_f_ = 0.47 (EtOAc/petroleum ether 7:3). ^1^H NMR (500 MHz, CDCl_3_) = δ: 0.73 (t, *J* = 5.9 Hz, 2H), 1.35 (t, *J* = 5.9 Hz, 2H), 2.33 (s, 3H), 2.43 (t, *J* = 5.4 Hz, 2H), 4.33 (t, *J* = 5.4 Hz, 2H), 5.22 (bs, 1H), 6.88 (s, 1H), 6.98 (t, *J* = 7.6 Hz, 1H), 7.10 (t, *J* = 7.6 Hz, 1H), 7.51 (d, *J* = 8.3 Hz, 1H), 7.67 (d, *J* = 8.0 Hz, 1H) ppm; ^13^C NMR (125 MHz, CDCl_3_) = δ: 11.98, 16.41, 20.34, 38.88, 67.53, 109.51, 112.62, 119.20, 120.15, 121.39, 125.40, 128.94, 131.03, 135.72, 171.28. Elemental analysis: calcd for C_16_H_17_N_3_O: C, 71.89; H, 6.41; N, 15.72; found: C 71.99, H 6.62, N 15.64.

*N*-(1-Cyanocyclopropyl)-3-(3-methyl-1*H*-indole-1-yl)propanamide **(7b)**

According to the same procedure described for **1a**, a solution of 3-(3-methyl-1*H*-indol-1-yl)propanoic acid **20b** (50 mg, 0.25 mmol) was reacted with 1-aminocyclopropanocarbonitrile **23** (35 mg, 0.30 mmol) and a crude residue was obtained. It was purified by silica gel column chromatography using EtOAc/petroleum ether 6:4 to obtain the pure coupling product **7b** (34.7 mg, 52%); consistency: light orange powder; M.p.: 114–115 °C, *R*_f_ = 0.59 (EtOAc/petroleum ether 6:4). ^1^H NMR (500 MHz, CDCl_3_) = δ: 0.78 (t, *J* = 6.1 Hz, 2H), 1.34 (t, *J* = 6.1 Hz, 2H), 2.29 (s, 3H), 2.54 (t, *J* = 5.5 Hz, 2H), 4.40 (t, *J* = 5.5 Hz, 2H), 5.55 (bs, 1H), 6.84 (s, 1H), 7.11 (t, *J* = 6.9 Hz, 1H), 7.20 (t, *J* = 7.8 Hz, 1H), 7.25 (s, 1H), 7.55 (d, *J* = 7.8 Hz, 1H) ppm; ^13^C NMR (125 MHz, CDCl_3_) = δ: 9.48, 16.52, 20.18, 37.04, 42.02, 108.95, 111.00, 119.01, 119.24, 119.63, 121.77, 125.64, 128.91, 135.69, 171.26. Elemental analysis: calcd for C_16_H_17_N_3_O: C, 71.89; H, 6.41; N, 15.72; found: C 71.77, H 6.32, N 15.77.

*N*-(1-Cyanocyclopropyl)-3-(5-fluoro-1*H*-indol-1-yl)propanamide **(8b)**

According to the same procedure described for **1a**, a solution of 3-(5-fluoro-1*H*-indol-1-yl) propanoic acid **21b** (50 mg, 0.24 mmol) was reacted with 1-aminocyclopropanocarbonitrile **23** (34.33 mg, 0.29 mmol) and a crude residue was obtained. It was purified by silica gel column chromatography using EtOAc/petroleum ether 7:3 to obtain the pure coupling product **8b** (22.5 mg, 28%); consistency: white powder; M.p.: 120–122 °C, *R*_f_ = 0.63 (EtOAc/petroleum ether 7:3). ^1^H NMR (500 MHz, CDCl_3_) = δ: 0.89 (t, *J* = 6.0 Hz, 2H), 1.38 (t, *J* = 6.0 Hz, 2H), 2.58 (t, *J* = 6.2 Hz, 2H), 4.45 (t, *J* = 6.2 Hz, 2H), 5.79 (bs, 1H), 6.42 (s, 1H), 6.95 (t, *J* = 9.0 Hz, 1H), 7.11 (s, 1H), 7.20–7.26 (m, 2H) ppm; ^13^C NMR (125 MHz, CDCl_3_) = δ: 16.75, 20.45, 37.05, 42.56, 101.94 (d, *J* = 4.6 Hz), 105.94, 106.13, 109.88 (d, *J* = 9.9 Hz), 110.35 (d, *J* = 26.4 Hz), 119.69, 129.81, 132.19, 158.05 (d, *J* = 235.0 Hz), 171.10. Elemental analysis: calcd for C_15_H_14_FN_3_O: C, 66.41; H, 5.20; N, 15.49; found: C 66.32, H 5.36, N 15.66.

### 3.2. Enzyme Assays against Rhodesain

Preliminary screening with rhodesain was performed with 100 µM inhibitor concentrations using an equivalent amount of DMSO as negative control. Rhodesain was recombinantly expressed as previously described [34]. Product release from substrate hydrolysis (Cbz-Phe-Arg-AMC, 10 µM) was determined continuously over a period of 10 min. Compounds showing at least 50% inhibition at 100 µM were subjected to detailed assays. The assay buffer contained 50 mM sodium acetate, pH = 5.5, 5 mM EDTA, 200 mM NaCl, and 0.005% Brij. The enzyme buffer contained 5 mM DTT rather than Brij. Product formation was monitored continuously for 10 min at room temperature. Inhibitor solutions were prepared from stocks in DMSO. Each independent assay was performed twice in duplicate in 96-well plates in a total volume of 200 µL. Fluorescence of the product AMC of the substrate hydrolyses was measured using an Infinite 200 PRO microplate reader (Tecan, Männedorf, Switzerland) at room temperature with a 380 nm excitation filter and a 460 nm emission filter. The dissociation constants *K*_i_ were obtained from progress curves (10 min) at various concentrations of inhibitor by fitting the progress curves to the 4-parameter IC_50_ equation:$$y=\frac{y_{max}-y_{min}}{1+{\left(\frac{\left[I\right]}{{IC}_{50}}\right)}^{S}}+y_{min}$$
where y [dF/min] was the substrate hydrolysis rate, y_max_ was the maximum value of the dose-response curve that is observed at very low inhibitor concentrations, y_min_ was the minimum value that was obtained at high inhibitor concentrations, and s denoted the Hill coefficient, *K*_i_ = IC_50_/(1 + [S] K_m_^−1^).

### 3.3. FP-2 Inhibition

Compounds were tested for inhibition of recombinant FP-2, which was expressed and purified as previously reported [37]. Stock solutions of the compounds, substrate, and the positive control E64 (Sigma Aldrich, St. Louis, MO, USA) were prepared at 10 mM in 100% DMSO. The compounds were incubated in 96-well white flat-bottom plates with 30 nM recombinant FP-2 for 10 min at room temperature in an assay buffer of 100 mM sodium acetate, pH 5.5, with 5 mM DTT. After incubation, the fluorogenic substrate Cbz-Leu-Arg-AMC (R&D Systems, Minneapolis, MN, USA) was added at a concentration of 25 µM in a final assay volume of 200 µL. Fluorescence was monitored with a Varioskan Flash (ThermoScientific, Waltham, MA, USA) with excitation 355 nm and emission 460 nm. The IC_50_ values were calculated using GraphPad Prism v. 6 (GraphPad Software Inc., San Diego, CA, USA)) based on a sigmoidal dose response curves.

### 3.4. Antitrypanosomal Activity

IC_50_ values for the antitrypanosomal activity of compounds (**1a**, **2a**, **2b**, **3a**, **6b**) were determined using the ATPlite assay as described previously [38,39]. The final compound concentrations in the microplates were 33.3 µM to 65 nM.

### 3.5. Docking Experiments

AutoDock4 (AD4) was employed for the molecular docking calculations [40]. To model the formation of the covalent adduct between the ligand and the protein, the “flexible side chain method” was used [41]. In the preparation phase, the ligand was modeled within the Maestro suite to incorporate two additional atoms (a sulfur and a carbon atom) to be aligned with the C25 residue of the protein. The X-ray structure of rhodesain (PDB code: 2P86) underwent preliminary adjustments for docking purposes by using the Protein Preparation Wizard integrated into the Schrödinger suite [42]. Alignment of the ligand with the reactive cysteine was facilitated by scripts available from the AD4 website. Protein grid maps, employing ligand atom types as probes, were generated with AutoGrid4 (v 4.2.6) software. The dimensions of the enzyme grid box were set at 60 Å × 60 Å × 60 Å with a 0.375 Å spacing, centered on the coordinates of the native co-crystallized ligand. The docking calculations were executed with flexibility allowed for the modified cysteine/ligand residue. The Lamarckian genetic algorithm (LGA) was used for these docking simulations, encompassing 100 LGA runs. All other settings were maintained at their default values. The docking results were subsequently grouped based on the root mean square deviation (RMSD) criterion, whereby solutions differing by less than 2.0 Å were considered part of the same cluster. The ranking of these clusters was determined based on the calculated lowest free energy of binding (ΔGAD4). Visualization of the results was carried out using the UCSF Chimera X (version 1.7.1) software [43].

### 3.6. Molecular Dynamics Experiments

The ligand-protein complex for **2b** obtained from the docking experiments was used to construct a molecular dynamics system that was solvated in water within an orthorhombic box with a buffer distance of 10 Å. Additionally, the overall charge of the system was automatically neutralized by Maestro’s System Builder utility [44]. The salt concentration was set to 0.15 M NaCl. The chosen force field for constructing the system was OPLS3 [45].

The Desmond module within the Schrodinger suite was used for a 500 ns long MD simulation. NPT was chosen as the experimental environment (constant pressure of 1.01325 bar and constant temperature of 300 K). In detail, the temperature was controlled using the Nosé–Hoover thermostat with a 1.0 ps relaxation time [46,47]. The pressure was controlled using the Martyna–Tobias–Klein barostat with an isotropic coupling style and a 2.0 ps relaxation time [48,49]. The cutoff distance for short-range nonbonded interactions was 9 Å. To minimize the computation time, nonbonded forces were calculated using a RESPA integrator where the short-range forces were updated every two steps [50], and the long-range forces were updated every six steps. The Desmond Simulation Interaction Diagram tool was used to analyze the receptor–ligand interactions during the entire MD trajectory.

## 4. Conclusions

In conclusion, in this article we focused our attention on the development of novel rhodesain and FP-2 inhibitors for the treatment of NTDs and PRDs. Our experimental studies have identified a lead-like compound, i.e., inhibitor **2b**, with a *K*_i_ of value of 5.06 µM towards rhodesain and an IC_50_ = 40.43 µM against FP-2. This novel nitrile is a promising antitrypanosomal and antimalarial agent worthy of further development.

## Data Availability

Data will be made available on request to the corresponding author.

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
