# Peer review of "Development of Novel Peptidyl Nitriles Targeting Rhodesain and Falcipain-2 for the Treatment of Sleeping Sickness and Malaria"

_ijms, 2024, doi:10.3390/ijms25084410_

Round 1
Reviewer 1 Report
Comments and Suggestions for Authors
See attached

Comments on the Quality of English LanguageAuthor Response
Attached

Reviewer 2 Report
Comments and Suggestions for Authors
The manuscript ijms-2953199 reports the design, synthesis, and biological evaluation of novel heterocyclic compounds. The manuscript is well written, and I recommend the publication after major revisions as follows below:
1) Provide as Supplementary Material all spectra used to determine the chemical structure of the synthetic compounds.
2) In the Supplementary Material provide the method used to determine the purity of the compounds used in the biological assays. Do the authors used NMR with a standard compound to determine the purity percentage or other method? Please, explain it in the materials and methods section.
3) Please, provide the melting point values for the synthetic compounds (section 3.1.).
4) Please, provide the mass spectra of the synthetic compounds as Supplementary Materials and the corresponding signals in the section 3.1.
5) Molecular dynamics simulation must be carried out at least in triplicate to verify the consistency and reproducibility of the simulation and trend. Please, do it and compare the triplicate with the plots of RMSD and RMSF.
6) Provide the positive controls of the antitrypanosomal assays.
7) The authors claimed the use of the positive control E64 in the FP-2 inhibition assays, however, where is the plot and value?
8) What is the reason that in Figure 2 the authors only provided the antitrypanosomal activity of compound 2b and not all the assayed compounds? Please, provide the antitrypanosomal plots for the compounds 1a, 2a, 3a, and 6b. This is important to verify the potency and efficacy.
Round 2
Reviewer 2 Report
Comments and Suggestions for Authors
The authors of the manuscript ijms-2953199 replied to the Reviewers’ questions and improved the quality of the work. Thus, I recommend the publication of this work.